# SPONGE: Competing Sparse Language Representations for Effective Cross-Lingual Knowledge Transfer in Healthcare

**Jens-Michalis Papaioannou**[*]                                        *michalis.papaioannou@bht-berlin.de*
*Berlin University of Applied Sciences and Technology*
*Leibniz University Hannover*

**Alexei Figueroa Rosero**[*]                                        *alexei.figueroarosero@bht-berlin.de*
*Berlin University of Applied Sciences and Technology*
*Leibniz University Hannover*

**Conor Fallon**                                        *cfallon@bht-berlin.de*
*Berlin University of Applied Sciences and Technology*

**Anna Capilla**                                        *annacapilla@gmail.com*
*Independent Researcher, Berlin, Germany*

**Alexandra Bekiaridou**                                        *abekiaridou@northwell.edu*
*Feinstein Institutes for Medical Research, Northwell Health*

**Stavros Zanos**                                        *szanos@northwell.edu*
*Feinstein Institutes for Medical Research, Northwell Health*

**Wolfgang Nejdl**                                        *nejdl@l3s.de*
*Leibniz University Hannover*

**Alexander Löser**                                        *aloeser@bht-berlin.de*
*Berlin University of Applied Sciences and Technology*

**Reviewed on OpenReview:** *https://openreview.net/forum?id=OevFdPgk3h*

[*]Equal contribution.

## Abstract

In domains with privacy constraints, most knowledge resides in siloed datasets, hindering the development of a model with *all* relevant knowledge for a task. Clinical NLP is a prime example of these constraints in practice. Research in this area typically falls back to the canonical setting of sequential transfer learning, where a model pre-trained on large corpora is finetuned on a smaller annotated dataset. An avenue for knowledge transfer among diverse clinics is *multi-step sequential transfer learning* since models[1] are more likely to be shared than private clinical data. This setting poses challenges of cross-linguality, domain diversity, and varying label distributions which undermine generalisation. We propose SPONGE, an efficient prototypical architecture that leverages competing sparse language representations. These encompass distributed knowledge and create the necessary level of redundancy for effective transfer learning across multiple datasets. We identify that prototypical classifiers

---

[1]Huggingface biomedical models

are critically sensitive to label-recency bias which we mitigate with a novel strategy at inference time. SPONGE in combination with this strategy significantly boosts generalisation performance to unseen data. With the help of medical professionals, we show that the explainability of our models is clinically relevant. We make all source code[2] available.

## 1 Introduction

In real-world machine learning applications, access to labeled data is often limited, whether due to intellectual property reasons or privacy constraints. Clinical Natural Language Processing (NLP) tasks exemplify this challenge, where these limitations hinder the unification of all medical knowledge. A common approach to handling decentralised and private data is Federated Learning (FL). However, FL performance suffers when data are non-i.i.d. (Nguyen et al., 2023), which is prevalent for most of the clinically relevant outcome classes (diagnoses, medications, procedures). Sequential transfer learning (Howard & Ruder, 2018) is a well-established and simple alternative to FL when model checkpoints, rather than data, are used for knowledge transfer. Furthermore, this is particularly relevant for the case of NLP models of the BERT family, since attacks on extracting their training data are significantly less successful than with generative models (GPT family)(Lehman et al., 2021; Vakili & Dalianis, 2021; Huang et al., 2022), therefore preserving privacy better. Sequential transfer learning is also the de facto standard for tasks with limited labeled data, usually involving a two-step approach: pretraining on large data corpora and fine-tuning on a smaller domain-specific dataset. Nevertheless, more than two steps (datasets) have also proven beneficial for knowledge transfer (Poth et al., 2021; Jeong et al., 2020).

At the core of clinical data are Electronic Health Records (EHR), which document the medical history of patients. Upon patient admission, predictions of diagnoses, procedures, medications, and mortality risk help optimize resource allocation in clinical facilities. Clinical Decision Support Systems (CDSS) increasingly rely on NLP methods to assist medical professionals. Models from various clinics and specialties have been made available by researchers in large hosting platforms like *Huggingface* (Wolf et al., 2020). For example, clinical BERT models trained by Zhongshan and Qingpu Hospitals[3] (Wang et al., 2023) or Charite Hospital[4] (Bressem et al., 2024) among others are publicly available and can be finetuned by other clinical facilities for downstream tasks.

In diagnoses prediction state-of-the-art (SOTA) methods augment Transformer representations (BioMed-BERT (Tinn et al., 2023)) with prototypical networks (Figueroa et al., 2024; van Aken et al., 2022). Diagnoses prediction generally poses significant challenges: 1) In practice diagnoses occur with widely varying prevalence, exhibiting a pronounced long-tail distribution (Papaioannou et al., 2022); 2) Medical knowledge and data are unevenly distributed across regions and languages. For example, the higher incidence of gastric cancer in Japan (Naylor et al., 2006) or Chagas disease in South America (Martins-Melo et al., 2014); 3) Privacy concerns isolate patient data across medical facilities, hindering unified model training. These challenges are amplified by the fact that Transformer models are usually pretrained on large corpora in high-resource languages (mainly English) limiting performance in low-resource languages.

Sequential transfer learning addresses the constraints of clinical data, namely privacy and data isolation, uneven diagnosis distribution, low-resource settings, and multilinguality. Multiple fine-tuning steps have also been shown to improve downstream performance on low-resource diagnoses prediction (Papaioannou et al., 2022). Research has centered on finding optimal sequences of tasks (Lim et al., 2024; Poth et al., 2021), highlighting the performance sensitivity of Transformers to the task order. However, finding a training order for optimal knowledge transfer requires access to all datasets, which does not comply with the clinical data restrictions mentioned above. We argue that creating robust solutions subject to these constraints involves 1) A realistic simulation of knowledge transfer that reflects the challenges of working with clinical data, 2) An architecture that is *agnostic to the fine-tuning sequence.*

---

[2]**Download** the repository at `https://anonymous.4open.science/r/HSPONGE-6DDD`
[3]BERT from Zhongshan and Qingpu Hospitals.
[4]BERT from Charite Hospital.

We gather a collection of six clinical datasets encompassing different clinics, writing styles, languages, number of patients, and diagnoses distributions. In Figure 1 we illustrate the label spaces of our five training datasets, which we complement with an additional dataset used only for testing. We train and evaluate SOTA architectures strictly following clinical data constraints, i.e. using model checkpoints rather than data as the medium of knowledge transfer (see Figure 1 right). Additionally, we focus on assessing *sequence-agnostic* performance, reflecting the realistic scenario in which clinics are unaware of the data seen by publicly available models.

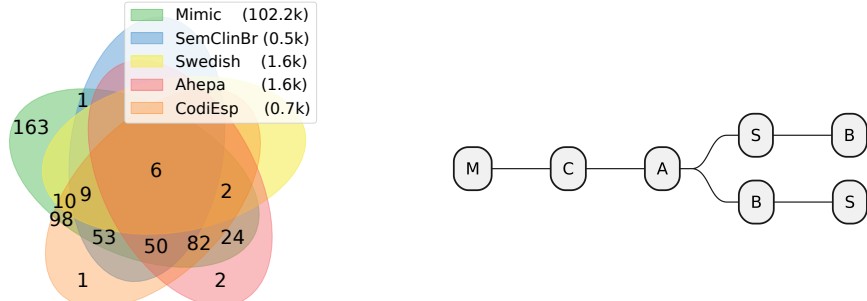

Figure 1: **(Left)** Label space relationships across five clinical datasets used for training. *Legend*: number of training samples for each dataset. **(Right)** Six (two-steps or more) training sequences are generated from two 5-step permutations of the datasets that start with MIMIC ($M$). Each node is the end of a sequence and results in a model checkpoint; e.g, $C$ indicates the model checkpoint for $M \rightarrow C$. In our experiments, we ensure that the model checkpoints (and not data) are the only medium of knowledge transfer.

We find that SOTA models for diagnoses prediction such as S-Proto (Figueroa et al., 2024) struggle under realistic clinical conditions. We argue that building non-interfering knowledge redundancy during *sequential fine-tuning* is key for knowledge transfer in this scenario. However, deep neural networks are usually dense, i.e., they utilize all parameters during prediction. In sequential settings, deep neural networks often suffer from knowledge overwriting, which impairs effective transfer (Ling et al., 2024).

We introduce SPONGE: an efficient architecture that boosts knowledge transfer in diagnoses prediction by creating end-to-end sparse language representations. We construct these by integrating concepts from parameter-efficient fine-tuning (PEFT) (Pfeiffer et al., 2020; Poth et al., 2023), prototypical networks (Snell et al., 2017), and conditional computation (Bengio et al., 2013) using a non-parametric *winner-takes-all* (Yuille & Grzywacz, 1989) mechanism. Inspired by biological neural circuits, the sparse subnetworks in SPONGE are full-fledged predictors that compete to predict a diagnosis. While requiring only ≈2.6% of the trainable parameters, our method outperforms current SOTA in diagnoses prediction. We validate this extensively by analyzing both single and multi-step transfer learning scenarios.

Catastrophic forgetting (McCloskey & Cohen, 1989) during sequential fine-tuning impairs generalization. We find that prototypical classifiers tend to focus disproportionately on *recent* labels when compared to traditional Transformers, reducing generalization to unseen datasets. However, explicitly preventing catastrophic forgetting can degrade performance on target tasks (Pfeiffer et al., 2020). We propose an alternative approach: $\mathcal{H}$ydra, a strategy that improves generalization to unseen datasets without modifying SPONGE's architecture. $\mathcal{H}$ydra efficiently improves macro AUROC and PRAUC by 7.5 and 3.5 percentage points, respectively, with a negligible increase in inference parameters. SPONGE and $\mathcal{H}$ydra function as adapters that can be loaded and fine-tuned by clinics as needed to boost knowledge transfer.

Additionally, our method inherits the explainability properties of previous SOTA (Figueroa et al., 2024) by predicting in a latent prototypical space (Chen et al., 2019). With input from medical professionals, we validate that the observed improvement in knowledge transfer is underpinned by clinically meaningful explanations (van Aken et al., 2022).

**Contributions**

- To our knowledge, our evaluation of multi-step sequential transfer learning is the largest to date in simulating knowledge transfer for diagnoses prediction with real-world clinical data constraints.

- We propose a novel architecture of sparse competing subnetworks that achieves superior knowledge transfer in diagnoses prediction.

- Our analysis identifies weaknesses in current SOTA approaches, which we address with an efficient strategy that mitigates catastrophic forgetting and improves generalization to unseen datasets.

## 2 Related Work

**Diagnoses prediction from textual data.** Transformer models have been widely applied to diagnoses prediction (He et al., 2025; Roehr et al., 2024; van Aken et al., 2021; Rasmy et al., 2021). Yang et al. (2022) employ prompt tuning combined with a contrastive loss to improve few-shot prediction of diagnoses codes, while Liu et al. (2021) propose a convolutional attention network that captures multi-scale representations and integrates cross-entropy with focal loss to better handle rare labels. Zhang et al. (2022) incorporate structural information from medical documents and reconcile code embeddings to address class-imbalance and stylistic heterogeneity in clinical texts. The work most closely related to ours, S-Proto (Figueroa et al., 2024) improves diagnoses prediction, particularly for rare codes, by augmenting a Transformer with a sparse prototypical network. The authors attribute the improved performance to the competing subnetworks in their prototypical layer. In contrast, we construct competing end-to-end subnetworks that integrate both the prototypical network and the Transformer to boost diagnoses prediction for high and low-resource datasets.

**Cross-lingual transfer** is a popular topic in NLP (Hämmerl et al., 2024; Philippy et al., 2023). Most research has focused on identifying optimal dataset sequences and configurations, extracting dataset properties that enhance performance on downstream tasks (Lim et al., 2024; Protasov et al., 2024; Lin et al., 2019; Malkin et al., 2022). Given clinical data constraints, Papaioannou et al. (2022) find that sequential training for diagnoses prediction boosts downstream performance for low-resource datasets, however, only for optimal dataset sequences. In contrast, we propose a robust architecture that enables cross-lingual transfer *independently* of sequence order, and the dataset properties; i.e., size, labels, distribution, and language.

**Parameter efficient fine-tuning (PEFT) and generalization.** Liu et al. (2024); Lialin et al. (2023); Chalkidis et al. (2021) highlight the importance of PEFT using adapters to improve zero-shot classification performance. In our work, adapters play a crucial role in inducing sparsity. We use a pool of adapters to generate multiple Transformer representations, which compete via a *winner-takes-all* (Yuille & Grzywacz, 1989) mechanism. Additionally, PEFT enables our model to scale the number of subnetworks efficiently.

**Parameter selective training** approaches have centered on using masks to update specific parameters (Somayajula et al., 2024; Sung et al., 2021; Winter et al., 2022). Similarly, we selectively update only a subset of model parameters. However, rather than applying weight level updates, we constrain updates to subnetworks responsible for end-to-end representations. Moreover, Ansell et al. (2022) use sparse fine-tuning to boost cross-lingual transfer using prior knowledge of the target task and language. In contrast, our subnetwork selection strategy operates without prior knowledge of downstream tasks.

**Subnetwork routing and lateral inhibition.** With ever-growing parameters in Transformers, a Mixture of Experts (MoE) offers an efficient path for scaling computation (Shazeer et al., 2017; Lepikhin et al., 2021; Fedus et al., 2022; Du et al., 2022; Chi et al., 2022). Our approach differs from MoE in that 1) it does not fragment the input (tokens) 2) we use no additional routing loss functions or gating mechanisms 3) our subnetworks are independent predictors that *compete* to produce the best representation. In fact, we leverage conditional computation (Bengio et al., 2013), and are closer to single-layer biologically inspired neural networks (Figueroa et al., 2025; Bricken et al., 2023; Liang et al., 2021) to learn sparse representations through lateral inhibition. However, we extend this to deeper architectures integrating Transformer models with prototypical classification in a latent metric space.

**Sequential learning and catastrophic forgetting.** Sequential learning presents significant challenges for deep neural networks. A well-established trade-off in this setting is the *stability-plasticity* dilemma, which is central to continual learning (Biesialska et al., 2020; Huang et al., 2021). In cross-lingual transfer, explicitly

mitigating catastrophic forgetting (i.e., optimizing for stability) may result in reduced downstream performance (Pfeiffer et al., 2020). We use *winner-takes-all* as a non-parametric inhibition approach that creates non-interfering parallel representations, improving both plasticity and stability. However, our objective is to maximize knowledge transfer under clinical data constraints; hence, we evaluate downstream performance on low-resource datasets of clinics.

## 3 Multilingual Clinical Datasets

We train and test our models on sequences constructed from five different clinical datasets across multiple languages (see Figure 1, right). Additionally, we evaluate their generalization performance on a multilingual dataset with the same EHRs in seven different languages in a zero-shot setting. For all datasets, we map the diagnoses to the Clinical Classifications Software Refined (CCSR)[5] code space. CCSR offers advantages over International Classification of Diseases (ICD-10), including improved clinical relevance and simplified categorization.

**MIMIC-IV (M)** (Johnson et al., 2021; 2023) is an *English* clinical dataset containing cases from the Intensive Care Unit (ICU) and other hospital departments. Admission and discharge notes are simulated from the EHRs as in van Aken et al. (2021).We use only the ICD-10 subset, which we map to the corresponding CCSR categories.

**CodiEsp (C)** contains clinical case studies in *Spanish*, with patients from various medical specialties, including oncology, urology, cardiology, pneumology, and infectious diseases (Miranda-Escalada et al., 2020).

**AHEPA-Cardio (A)** are cardiology discharge summaries in *Greek* (Papaioannou et al., 2022).

**SemClinBr (B)** comprises *Portuguese* clinical notes from multiple Brazilian medical institutions across various specialties (Oliveira et al., 2022).

**Stockholm University - Gastrointestinal (S)**(Lamproudis et al., 2023) consists of EHRs from a gastroenterology department. These are in *Swedish* and sourced from the Swedish Health Record Research Bank (Dalianis et al., 2015)[6].

**Zero-Shot Datasets: DisTEMIST.** We use the dataset of Miranda-Escalada et al. (2022) for zero-shot evaluation. It comprises the same EHRs expressed in *Spanish*, *English*, *Catalan*, *Portuguese*, *French*, *Italian*, and *Romanian*.

**Data Splits and Label Space.** For all datasets, we use stratified sampling (Sechidis et al., 2011) to create train, validation, and test splits (see Table 5 in Appendix A). We remove EHRs with less than four labels. Figure 1 summarizes the label space and training sample counts.

## 4 Sequential Transfer Learning for Diagnoses prediction

**Diagnoses prediction** is a multilabel classification task with a large label space, in our case, comprising 509 CCSR codes (98% of the full CCSR specification). We focus on knowledge transfer for this task using only model checkpoints for transfer to simulate real-world clinical constraints. Accordingly, we adopt the canonical sequential transfer learning setup involving both single and multi-step fine-tuning (Ruder, 2019). This involves sequential training with all permutations of our datasets, evaluating performance **independently** of any specific fine-tuning order.

We initialize all architectures on this task with MIMIC, our high-resource dataset. Next, we construct all permutations of the four low-resource datasets to generate sequence variations. This process results in $64+1$ distinct sequences (including the one-step sequence on MIMIC, see Appendix E). Figure 1 (*right*) illustrates two example sequences.

---

[5]CCSR categories

[6]This research has been approved by the Swedish Ethical Review Authority under permission no.2019-05679. The use of Stockholm EPR Gastro ICD-10 Pseudo Corpus II with the amendment no 2022-02386-02.

**Evaluation.** The goal of our experiments is to assess how effectively our architecture transfers relevant knowledge and generalizes to unseen data at each step of the sequence. For example, sequences involving one training dataset are evaluated on four unseen test datasets, those trained on two evaluated on three, and so on. DisTEMIST is used exclusively as a test set. We evaluate all resulting model checkpoints $(64 + 1)$, measuring averages of macro AUROC, micro AUROC, and macro PRAUC to capture *sequence agnostic* scores.

**MIMIC (High-resource).** We report results on this dataset as it serves as the standard benchmark for diagnosis prediction (Roehr et al., 2024; van Aken et al., 2021). This evaluation corresponds to the one-step sequential transfer learning scenario, i.e., the sole single-dataset sequence in our experiments.

**Final datasets (Low-resource).** We rank architectures based on the performance on the *final dataset* in each sequence. This evaluation measures downstream performance independently of the datasets seen by the model during sequential training. This is consistent with the performance expected by clinics when fine-tuning models without explicit knowledge of their prior training data. In this evaluation, we average all performances in all sequence configurations. We evaluate 64 sequences of two datasets or more, with a total of 16 sequences for every target dataset (see Appendix E).

**Zero-shot generalization.** In contrast to the standard *zero-shot cross-lingual transfer setting* (Hu et al., 2020), we focus on models trained sequentially on multiple datasets. This implies that variations in data distributions, label spaces, and language typologies may contribute or interfere with every additional training dataset. We probe the generalization capabilities of all models on DisTEMIST in addition to all unseen datasets. We evaluate $64 + 1$ checkpoints per model and report the average for each metric for 1) all unseen datasets, 2) unseen languages within DisTEMIST, and 3) seen languages within DisTEMIST.

## 5 Methods

Several works have tackled diagnoses prediction with Transformer networks and architectural augmentations. van Aken et al. (2022) incorporate prototypical classifiers to use latent metric spaces, enhancing both the explainability and performance of model predictions. Figueroa et al. (2024) extend this approach in S-Proto by adding sparsity to the prototypical classification layer using a *winner-takes-all* mechanism. Specifically, S-Proto leverages competing subnetworks to model distinct phenotypes of diagnoses, further boosting performance. These architectural enhancements to Transformer representations achieve state-of-the-art performance in diagnoses prediction. However, they struggle when exposed to multi-step sequential transfer learning, since sequentially updating a single Transformer representation creates inter-task interference. We speculate that this is due to the competing subnetworks, which share a large number of parameters, leading to entangled representations. To address this, we propose extending sparsity throughout the entire network, rather than limiting it to the classification layer.

### 5.1 End-To-End Sparse Representation

The dense (non-sparse) modules of S-Proto (Figueroa et al., 2024) are the Transformer and the projection matrix to the prototypical space. We add competing subnetworks to these components to create end-to-end sparse representations, which is our main methodological contribution generalizing the work of Figueroa et al. (2024). This entails 1) creating in parallel multiple Transformer representations for the same input, 2) expanding the dimensionality of the projection layer to the prototypical space, and 3) selecting a *winner* subnetwork, hence, the *winner-takes-all* mechanism must be generalized for the entire architecture. We show an overview of the resulting architecture in Figure 2.

**Creating multiple Transformer representations** would require parallel embedding Transformers, which is computationally prohibitive. We opt for a simpler solution using multiple adapters (Figure 2 *left*), which significantly improves efficiency. Trainable adapters combined with a language model have shown competitive performance on classification tasks (Houlsby et al., 2019). We add $T$ task adapters (Pfeiffer et al., 2020) as modular components, freezing the rest of the Transformer while training.

$$TA^t(emb, r) = U^t(ReLU(D^t(emb))) + r \tag{1}$$

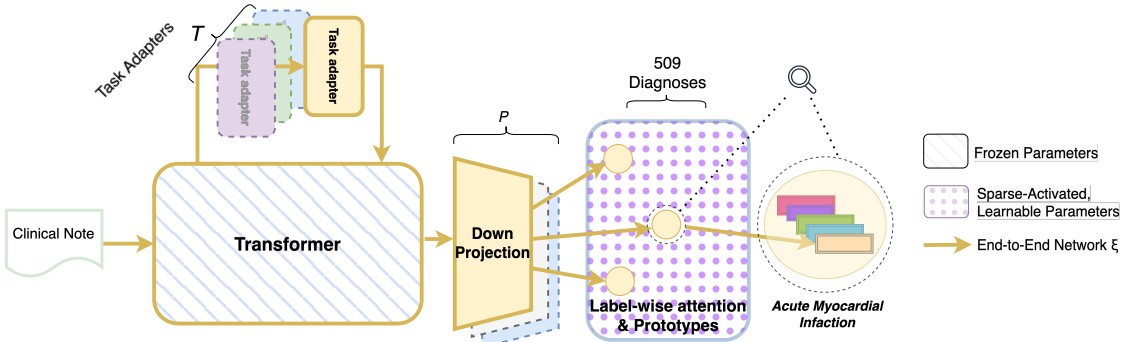

Figure 2: Classifying a clinical note with SPONGE: A pool of $T$ adapters (left) create multiple transformer representations. $P$ down-projection layers map these representations, which are then classified in a latent metric space with sparse label-wise attention. $\xi$ is the winner network among all competing subnetworks that map the input to the label space, resulting in sparse gradient updates.

$D^t$ and $U^t$ are the task adapter $t$ down and up projections, *emb* is the encoder representation and $r$ is the residual for a layer within the Transformer. We compute multiple token representations, one with every task adapter: $\eta^t \in \mathbb{R}^{E \times h}$, where $h$ is the hidden dimension of the encoder, and $E$ is the encoder sequence length.

**Expanding the dimensionality of the projection layer.** Additionally, we increase the number of down-projection layers $L \in \mathbb{R}^{h \times d}$ to $L \in \mathbb{R}^{P \times h \times d}$ (Figure 2 *center*), where $d$ is the hidden dimension and $P$ is the number of projection layers. We project $\eta^t$ which results in a representation $\psi^{t,p} = \langle \eta^t, L^p \rangle \in \mathbb{R}^{E \times d}$, where $L^p \in \mathbb{R}^{h \times d}$ is a single projection indexed by $p$, and $t \in \{1, ..., T\}$ and $p \in \{1, ..., P\}$ define a subnetwork. We refer to this as the *input subnetwork*, specified by $t$ and $p$, which enables an end-to-end sparse encoding of the EHR and is core to our contributions.

Let $\gamma \in \Gamma$ be an *output subnetwork* index in the prototypical layer, $c \in C$ be a specific class, therefore $\xi \in T \times P \times \Gamma$ defines the subnetwork that spans through the entire model and involves the combination of $t, p, \gamma$. We first map the token vectors $\psi^{t,p}$.

$$\phi^\xi = \langle \psi^{t,p}, W^\gamma \rangle^\top \in \mathbb{R}^{C \times E}$$

where $W^\gamma \in \mathbb{R}^{d \times C}$ are the labelwise-attention vectors. A class score $S^\xi = \text{softmax}(\phi^\xi)$ is computed for every token in the EHR. The mapped EHR, for a subnetwork $\xi$, in prototypical space is $v^\xi$.

$$v^\xi = \langle S^\xi, \psi^{t,p} \rangle \in \mathbb{R}^{C \times d}$$

SPONGE: **Generalizing *winner-takes-all*.** The Euclidean distance from a mapped EHR $v^\xi$ to a prototype $u^\xi$ is denoted by $\epsilon^\xi = \|u^\xi - v^\xi\|_2$ and the prediction of a subnetwork $\hat{y}^\xi = \sigma(-\epsilon^\xi)$, where $\sigma$ is the sigmoid function. We select the *winner* subnetwork $\overset{\star}{\xi}$ (Figure 2 *golden* path) with *the winner-takes-all* mechanism inhibiting all other subnetworks:

$$\hat{y} = \hat{y}^\xi \delta_{\xi, \overset{\star}{\xi}} \tag{2}$$

where $\delta$ is the Kronecker $\delta$ function. To compute the winning subnetwork $\overset{\star}{\xi}$ we start by selecting the *winner output subnetwork*:

$$\overset{\star}{\gamma}^{t,p} = \underset{\gamma}{\text{argmax}}(\hat{y}^\xi) \tag{3}$$

We use $\overset{\star}{\gamma}^{t,p}$ to select the *winner input subnetwork* $\overset{\star}{t}, \overset{\star}{p}$ with the mode over the classes for an EHR.

$$\overset{\star}{t}, \overset{\star}{p} = \underset{c}{\text{mode}}[\underset{t,p}{\text{argmax}}(\hat{y}^\xi \delta_{\gamma, \overset{\star}{\gamma}^{t,p}})] \tag{4}$$

Finally, the *end-to-end winner* subnetwork is:

$$\overset{\star}{\xi} = \underset{\gamma}{\mathrm{argmax}}(\hat{y}^{\overset{\star}{t},\overset{\star}{p},\gamma}) \tag{5}$$

We define the loss as the binary cross-entropy between $\hat{y}$ (see: Equation (2)) and the true labels $y$.

**Subnetwork exploration.** Training of sparse networks with *winner-takes-all* may result in *dead* subnetworks that are never selected and, thus, never updated. We modify the approach in (Bricken et al., 2023), inspired by $\epsilon - greedy$ (Sutton, 1995) exploration, to ensure broader subnetwork activation during training. At the start of training, we *explore* the top $k$ subnetworks $\xi$, choosing one of these randomly. During this phase, we anneal $k$ linearly until $k = 1$. Training proceeds with the top-1 subnetwork as in Equation (2). This strategy helps initialize and train all subnetworks, avoiding dead units, while still limiting computation by updating only one subnetwork per sample.

## 5.2 Boosting generalization

Prototypical networks use a latent distance to make predictions. Guided by the loss, these classifiers map tokens and shift prototypes in latent space. During sequential training, the learned prototypes become overly biased toward the *most recent* label distribution, severely restricting generalization in different label spaces. We argue that distributed knowledge still exists in the *input subnetworks* of SPONGE due to the overall sparsity (we examine this further in Section 8).

$\mathcal{H}$ydra: **exploiting distributed knowledge.** To counter the *label recency* bias, we store the prototypical parameters ($u$ and $W$) learned for each dataset in a training sequence, using them only for inference. Importantly, we only use parameters from past training steps for knowledge transfer, respecting the constraints of clinical data. Despite the growing number of *output subnetworks*, the *winner* subnetwork selection strategy of SPONGE remains applicable without requiring architectural changes. Although the number of parameters for inference increases after each dataset in a sequence, the increase is small w.r.t. the Transformer's size. In our experiments, in a five-dataset sequence, the added vectors amount to 6.6M parameters or 2.4% of the Transformer's 270M. A key advantage of the *winner-takes-all* selection is its compatibility with parameter accumulation, requiring no architectural changes nor added complexity like weight consolidation (Kirkpatrick et al., 2017) or knowledge distillation (Ermis et al., 2022).

$\mathcal{H}$SPONGE: $\mathcal{H}$ydra facilitates effective knowledge transfer to previously unseen data distributions. Importantly, SPONGE and $\mathcal{H}$ydra scale very well to larger Transformers since the dimensionality of the parameters is independent of their hidden size $h$. This approach supports modular scaling to arbitrary dataset sequences during *sequential training*. Furthermore, in the case of growing label spaces, this modularity would enable our models to train only the new parameters.

# 6 Experiments

Although generative large language models (LLMs) have shown strong performance in many language tasks, they still fall short of SOTA in many classification scenarios (Yang et al., 2024), particularly those with large label spaces. For diagnoses prediction, the best-performing approach remains fine-tuning Transformer representations with a classification head, for both encoder (Roehr et al., 2024; Figueroa et al., 2024) and decoder models (Gema et al., 2024), meanwhile decoder methods still fall short in this task (Grundmann et al., 2025).

We investigate how sparse subnetworks impact sequential transfer learning for diagnoses prediction. Therefore, we control for one Transformer architecture. We use *XLM-R* (Conneau et al., 2020) for all experiments, as it has been extensively studied for cross-lingual transfer (Philippy et al., 2023; Pfeiffer et al., 2020; Hu et al., 2020; Choi et al., 2020; Conneau et al., 2020). Although our methods are applicable to any Transformer architecture, we favor *XLM-R* over significantly larger Transformers, given the large number of experiments.

**Baselines.** We focus on the following architectures: 1) *XLM-R*, which is a cross-lingual model and the least sparse method (single dense network); 2) *XLM-R + A*, which extends the base model with a single task

adapter to ablate the effect of PEFT and an adapter in our architecture; and 3) S-Proto$_{\text{XLM-R}}$ (Figueroa et al., 2024), the current SOTA architecture for diagnoses prediction, which combines a dense Transformer with a sparse prototypical classifier. For the latter, we replace the BiomedBERT (Tinn et al., 2023) with XLM-R, since the former is not a cross-lingual Transformer.

**SPONGE Variations.** We evaluate multiple configurations of sparse subnetworks by varying the number of adapters and projection matrices. We name each variant SPONGE$^{T,P}$ where $T$ and $P$ stand for the number of task adapters and projections respectively. SPONGE$^{1,1}$ establishes the effect of having *no* sparsity in the encoder. SPONGE$^{1,6}$ displays the effect of having sparsity only in the projection stage before the prototypical layer. SPONGE$^{6,1}$ highlights the impact of no sparsity in the projection, but only in the Transformer using a pool of adapters. Given the effectiveness of adapters for cross-lingual transfer (He et al., 2021), our sparsest model, SPONGE$^{6,3}$, uses more adapters than projections.

All prototypical models (SPONGE and S-Proto$_{\text{XLM-R}}$) have five *output subnetworks* as in Figueroa et al. (2024). We detail hyperparameters in Appendix C.

## 7 Results

**Performance on MIMIC.** This dataset is a standard benchmark for NLP models in diagnoses prediction. We present the performance of all models in Table 1 (left). S-Proto$_{\text{XLM-R}}$ outperforms all dense baselines (lack subnetworks): XLM-R and XLM-R + A. Our variant with the largest number of subnetworks: SPONGE$^{6,3}$, outperforms all evaluated methods.

**Subnetwork exploration.** We show the impact of this strategy in the bottom section of Table 1 (left). The models SPONGE$^{1,1}$ and SPONGE$^{6,3}$ (marked with *-x*) indicate that they do *not* use *exploration*. In both cases, *exploration* improves PRAUC, benefiting the sparser SPONGE$^{6,3}$ more.

Table 1: (Left) model performance when trained solely on MIMIC. (Right) average performance on the last dataset of each sequence. SPONGE$^{6,3}$ consistently outperforms across both settings.

| Model | MIMIC Only | | | Sequential Training | | |
|---|---|---|---|---|---|---|
| | macro AUROC | micro AUROC | macro PRAUC | macro AUROC | micro AUROC | macro PRAUC |
| S-Proto$_{\text{XLM-R}}$ | 87.51 | 93.89 | 28.08 | 80.08 | 86.62 | 37.01 |
| SPONGE$^{1,1}$ | 89.68 | 94.67 | 33.18 | 87.47 | 91.78 | 52.07 |
| SPONGE$^{1,6}$ | 86.63 | 93.75 | 22.94 | 83.57 | 89.71 | 43.71 |
| SPONGE$^{6,1}$ | 86.55 | 90.42 | 31.42 | 86.19 | 90.71 | 48.93 |
| SPONGE$^{6,3}$ | **89.70** | **94.70** | **34.17** | **88.58** | **92.09** | **54.24** |
| XLM-R | 86.21 | 93.67 | 27.27 | 83.10 | 88.89 | 45.92 |
| XLM-R + A | 85.74 | 93.31 | 26.38 | 83.81 | 89.10 | 45.11 |
| SPONGE$^{1,1}$-x | 89.28 | 94.66 | 32.51 | – | – | – |
| SPONGE$^{6,3}$-x | 89.48 | 94.56 | 32.56 | – | – | – |

**Sequential training performance.** We show in Table 1 (right) the average performance of all models across the 64 training sequences (16 per target dataset; see Appendix E). XLM-R+A performs similarly to XLM-R, reinforcing the effectiveness of PEFT. S-Proto$_{\text{XLM-R}}$ performs the worst. This emphasizes the inter-task interference created by updating a single Transformer representation for sequential training. PEFT with *output subnetworks* shows significant gains. Specifically, 7.39 points in macro AUROC and 15.06 points in macro PRAUC when comparing SPONGE$^{1,1}$ to S-Proto$_{\text{XLM-R}}$.

Creating *input subnetworks* exclusively with either the adapter component or the projection is not as beneficial as creating them with both simultaneously. This is evident when comparing SPONGE$^{6,1}$ and SPONGE$^{1,6}$ against SPONGE$^{6,3}$. In general, increasing sparsity via more subnetworks boosts PRAUC, aligning with findings from Figueroa et al. (2024).

SPONGE$^{6,3}$ consistently demonstrates the best performance regardless of dataset order, language mix, label space diversity, or fine-tuning sequence length, while requiring $\approx 2.6\%$ of the training parameters of

XLM-R or S-PROTO$_{\text{XLM-R}}$(see Table 8 in Appendix F). We therefore focus on SPONGE[6,3] for all subsequent analyses.

**Downstream dataset performance.** We further analyze sequential training results by examining all permutations ending with each specific target dataset. We present this in Table 2 (top) and highlight how SPONGE[6,3] outperforms all methods for all metrics, showcasing the robustness to large distribution shifts (languages, dataset sizes, and labels). Generally, downstream dataset performance significantly benefits from multi-step sequential transfer learning, we detail this in Appendix B.

Although in practice clinical documentation is often decentralized and multilingual, we find that SPONGE also improves knowledge transfer in a standard monolingual transfer learning setting. We expand on this in Appendix G, where we reproduce the experiments of Section 6 for English translations of the datasets and SOTA English clinical Transformers.

**Number of training datasets.** We analyze the performance w.r.t. the number of datasets used in a training sequence. We present this in Table 2 (bottom). SPONGE[6,3] also significantly outperforms all other methods regardless of the number of datasets. We observe that updating all Transformer parameters (XLM-R, and S-PROTO$_{\text{XLM-R}}$) leads to performance degradation as the sequences increase in length.

Table 2: Classification performance per target dataset (top) and number of datasets (bottom) averaged over all permutations of sequences. SPONGE[6,3] outperforms all methods in both settings.

| Model/Dataset | macro AUROC | | | | micro AUROC | | | | macro PRAUC | | | |
|---|---|---|---|---|---|---|---|---|---|---|---|---|
| | A | B | C | S | A | B | C | S | A | B | C | S |
| S-PROTO$_{\text{XLM-R}}$ | 81.83 | 70.75 | 83.33 | 84.42 | 88.18 | 80.81 | 88.51 | 88.96 | 42.22 | 23.83 | 34.66 | 47.33 |
| SPONGE[6,3] | **90.05** | **82.65** | **91.03** | **90.62** | **94.18** | **86.68** | **93.97** | **93.53** | **59.93** | **42.16** | **55.22** | **59.67** |
| XLM-R | 85.78 | 73.30 | 85.67 | 87.63 | 91.39 | 81.78 | 89.70 | 92.69 | 58.85 | 27.73 | 39.20 | 57.90 |
| XLM-R+A | 84.83 | 75.93 | 86.38 | 88.08 | 90.27 | 83.05 | 90.33 | 92.76 | 55.65 | 28.54 | 38.77 | 57.52 |
| Model/ # datasets | 2 | 3 | 4 | 5 | 2 | 3 | 4 | 5 | 2 | 3 | 4 | 5 |
| S-PROTO$_{\text{XLM-R}}$ | 81.83 | 70.75 | 83.33 | 84.42 | 88.18 | 80.81 | 88.51 | 88.96 | 42.22 | 23.83 | 34.66 | 47.33 |
| SPONGE[6,3] | **90.05** | **82.65** | **91.03** | **90.62** | **94.18** | **86.68** | **93.97** | **93.53** | **59.93** | **42.16** | **55.22** | **59.67** |
| XLM-R | 85.78 | 73.30 | 85.67 | 87.63 | 91.39 | 81.78 | 89.70 | 92.69 | 58.85 | 27.73 | 39.20 | 57.90 |
| XLM-R+A | 84.83 | 75.93 | 86.38 | 88.08 | 90.27 | 83.05 | 90.33 | 92.76 | 55.65 | 28.54 | 38.77 | 57.52 |

**Performance on unseen data.** In Table 3 (left) we present the results for all unseen datasets: DisTEMIST and all datasets absent in a training sequence. Additionally, we adapt S-PROTO$_{\text{XLM-R}}$ and SPONGE with the $\mathcal{H}$ydra strategy. We denote this with $\mathcal{H}$ preceding the name of the model. S-PROTO$_{\text{XLM-R}}$ performs worse than XLM-R and XLM-R+A and fails to improve when employing $\mathcal{H}$ydra. SPONGE[6,3] performs better than XLM-R and XLM-R + A in PRAUC; however, it falls behind on AUROC. $\mathcal{H}$ydra proves effective, since $\mathcal{H}$SPONGE[6,3] outperforms all other methods.

**Performance on DisTEMIST.** The clinical notes of this dataset are multiple versions of the same EHR in different languages, therefore it is a good test bed for generalization. We distinguish between *unseen* and *seen* languages for all trained sequences. We present results in Table 3 center and right. Intuitively, for all models the performance on *seen* languages is higher than on *unseen* languages. $\mathcal{H}$SPONGE[6,3] consistently outperforms in both cases.

## 8 Analysis and Discussion

Our experiments show that multi-step sequential transfer learning is generally beneficial for diagnoses prediction. SPONGE outperforms all other methods on all target datasets, regardless of the sequence. Nevertheless, SPONGE struggles to generalize to unseen datasets, often underperforming XLM-R and XLM-R + A (Table 3). However, we demonstrate that $\mathcal{H}$ydra is an effective strategy to address this. We attribute the increased performance of $\mathcal{H}$SPONGE[6,3] to two main factors: 1) the sparsity and the modular architecture in SPONGE induced by *winner-takes-all*, which creates distributed Transformer representations; and 2) $\mathcal{H}$ydra's ability to leverage these representations, alleviating label recency bias.

Table 3: Average zero-shot performance of all sequences. *Left*: all datasets (DisTEMIST + those unseen during training). *Center, right*: DisTEMIST only, for both seen and unseen languages respectively.

| Model | Unseen datasets + DisTEMIST | | | DisTEMIST unseen languages | | | DisTEMIST seen languages | | |
|---|---|---|---|---|---|---|---|---|---|
| | macro AUROC | micro AUROC | macro PRAUC | macro AUROC | micro AUROC | macro PRAUC | macro AUROC | micro AUROC | macro PRAUC |
| S-Proto$_{\text{XLM-R}}$ | 68.77 | 65.40 | 12.58 | 69.36 | 65.11 | 10.91 | 72.69 | 67.23 | 14.97 |
| $\mathcal{H}$S-Proto$_{\text{XLM-R}}$ | 54.39 | 54.76 | 8.32 | 54.69 | 54.32 | 6.86 | 51.41 | 51.93 | 8.08 |
| SPONGE$^{6,3}$ | 74.46 | 71.15 | 20.91 | 74.44 | 70.69 | 19.27 | 77.74 | 73.23 | 23.47 |
| $\mathcal{H}$SPONGE$^{6,3}$ | **81.99** | **77.86** | **24.40** | **82.22** | **77.26** | **22.21** | **85.80** | **81.11** | **27.28** |
| XLM-R | 75.15 | 73.38 | 19.03 | 75.40 | 72.60 | 16.42 | 81.52 | 78.23 | 25.17 |
| XLM-R+A | 76.18 | 72.65 | 18.67 | 77.14 | 72.82 | 17.51 | 80.50 | 75.69 | 21.97 |

**Distributed Representations in SPONGE.** We investigate the existence of distributed knowledge after sequential training. For this, we test on the first dataset (MIMIC) after training on each of the 64 dataset sequences. As before, we report the average performance of all checkpoints which we denote as *Original* in Table 4. Additionally, we evaluate a variant of each checkpoint where we replace the classification layer with the one trained only on MIMIC; we denote this as $\mathcal{M}$. This allows the model's feature-building network to better align existing knowledge with the MIMIC label space. In other words, we decouple the Transformer representation from the classification layer and measure its impact. $\mathcal{M}$ favors the label space of MIMIC and thus mitigates *label recency* bias.

We observe that S-Proto$_{\text{XLM-R}}$ does not improve when employing $\mathcal{M}$. We argue that this degradation in performance is related to the dense projection $L$ which is crucial to map to the prototypical space. In contrast, XLM-R and XLM-R + A benefit from $\mathcal{M}$, hinting that the bare Transformer representations can recover knowledge from MIMIC, further emphasizing the need for sparsity in $L$ for S-Proto$_{\text{XLM-R}}$. Although SPONGE underperforms XLM-R and XLM-R + A, it outperforms all methods when employing $\mathcal{M}$. This significant boost in performance confirms that SPONGE retains MIMIC knowledge but is unable to access it once the prototypical parameters $u$ and $W$ have shifted after sequential-training. The $\mathcal{M}$ strategy is a simplification of $\mathcal{H}$ydra that explains how the latter enables access to distributed knowledge in the model for better generalization as seen in Table 3.

Table 4: Average performance on MIMIC of all 64 checkpoints. Every checkpoint has been trained on two or more datasets. *Original* denotes scores of the unmodified checkpoints; $\mathcal{M}$ corresponds to checkpoints modified by replacing their classification head with the one trained only on MIMIC; $\Delta$ denotes the difference.

| Model | Macro AUROC | | | Micro AUROC | | | Macro PRAUC | | |
|---|---|---|---|---|---|---|---|---|---|
| | Original | $\mathcal{M}$ | $\Delta$ | Original | $\mathcal{M}$ | $\Delta$ | Original | $\mathcal{M}$ | $\Delta$ |
| S-Proto$_{\text{XLM-R}}$ | 63.97 | 46.91 | -17.06 | 68.08 | 63.34 | -4.74 | 6.85 | 6.08 | -0.77 |
| SPONGE$^{6,3}$ | 68.43 | **82.30** | +13.87 | 79.62 | 83.50 | +3.88 | 14.58 | **20.39** | +5.81 |
| XLM-R | 76.87 | 78.91 | +2.04 | 80.74 | **87.48** | +6.74 | 16.68 | 18.53 | +1.85 |
| XLM-R+A | 73.00 | 78.79 | +5.79 | 76.68 | 87.04 | +10.36 | 11.27 | 17.40 | +6.13 |

**Winner-takes-all and optimal subnetwork.** We expand on the analysis of distributed knowledge by focusing on all subnetworks (full-fledged predictors) of SPONGE$^{6,3}$. For every $k^{th}$ subnetwork (90 in total) of this model, we compute all metrics which we present in Figure 3. The *Winner-takes-all* mechanism defines the order of the subnetworks indexed by $k$ (x-axis in the figure). For instance, $k = 89$ is the most inhibited subnetwork. In contrast, $k = 0$ is the single *winner* subnetwork that yields a prediction. We highlight with * the best-performing subnetwork, which proves to be within the top-18 for all metrics. Generally, given the aggregation in Equation (4), predictions are bundled in groups of 18 where performance is very similar, stemming from the 6 adapters $T$ and 3 projections $L$. This supports the idea that the inhibition mechanism in *winner-takes-all* produces multiple non-interfering predictors that are almost *optimal*.

**Hydra and label recency.** SPONGE$^{6,3}$ underperforms when not modified with $\mathcal{H}$ydra (see SPONGE$^{6,3}$ vs. XLM-R and XLM-R + A in Table 3), yet $\mathcal{H}$SPONGE$^{6,3}$ outperforms all methods on unseen data. We

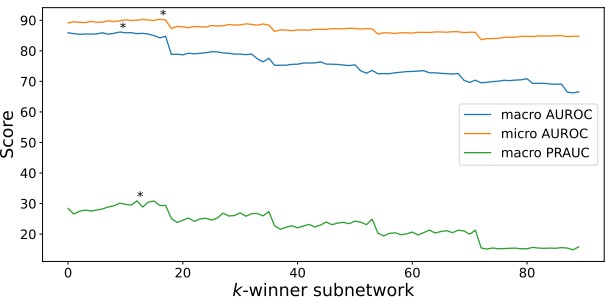

Figure 3: Scores achieved by SPONGE[6,3] for all subnetworks, ordered by the *winner-takes-all* selection mechanism (see Equation (5)). The decreasing nature of the curves highlights how the top activated subnetworks highly correlate with classification performance, reinforcing the suitability of this selection strategy. Asterisks ∗ point to the $k$ with the highest scores which are close to the *winner-takes-all* selection at $k = 0$.

investigate the impact of label recency on prototypical models when exposed to unseen data. Therefore, we compare $\mathcal{H}$SPONGE[6,3] and S-PROTO[XLM-R] since it is the SOTA architecture for this task. For a fair comparison to S-PROTO[XLM-R] we focus on its best-performing checkpoint on DisTEMIST i.e., the sequence M→B→C (see performance of S-PROTO[XLM-R] in Table 7 in Appendix D and Table 3). We analyze the performance of both models w.r.t. the *recency* of the labels for this sequence. For each label, we define *recency* as the proportion of the most recently trained samples (C) w.r.t all samples in M, B, and C.

**Recency and PRAUC.** We divide the labels into 8 logarithmically spaced *recency* groups, each containing at least 10 labels to ensure reliable estimation. Figure 4 (*bottom left*) shows the label distribution across these *recency* groups. In Figure 4 (*top left*) we present the difference($\Delta$) between the two models in macro PRAUC for the labels sorted by *recency* (see $x$-axis). $\mathcal{H}$SPONGE[6,3] performs significantly better on all but one label group, thus, $\Delta$ PRAUC is positive. This difference decreases as *recency* increases; notably, it is significantly larger for labels with lower *recency*. This illustrates the positive effect of $\mathcal{H}$ydra at mitigating the *recency bias*.

**Recency and explainability.** Prototypical methods create token saliencies that explain their predictions. Faithfulness quantifies how gradually masking the input according to these saliencies, deteriorates model performance. A lower faithfulness score indicates greater explainability (Atanasova et al., 2020). We generate token saliencies for $\mathcal{H}$SPONGE[6,3] and S-PROTO[XLM-R], and compute faithfulness for macro AUROC and macro PRAUC for the aforementioned *recency* groups. Figure 4 (*right*) presents the difference in faithfulness for $\mathcal{H}$SPONGE[6,3] and S-PROTO[XLM-R]. We expand these scores for all DisTEMIST languages (rows in the Figure). Similar to $\Delta$ PRAUC, the largest difference is in the first (least recent) label group. This relationship appears to hold independently of the language.

Both classification (PRAUC) and explainability (Faithfulness) metrics confirm that the prototypical layer alone (S-PROTO[XLM-R]) struggles to generalize in sequential training, likely due to an overemphasis on the most *recent* labels. However, $\mathcal{H}$SPONGE[6,3] overcomes this in both classification and explainability, reinforcing the importance of combining the distributed representations of SPONGE with $\mathcal{H}$ydra.

**Analysis of saliencies with doctors.** We further examine 30 patients and five diagnoses with the help of medical doctors. We compare the saliencies of S-PROTO[XLM-R] and $\mathcal{H}$SPONGE[6,3] for the previously analyzed sequence M→B→C. We aim to understand how the performance gap between the two models manifests in qualitative differences from a domain expert's perspective. We focus on testing AHEPA (A), which is not included in the sequence and contains the typologically most distant language (Greek). observe the same performance gap in faithfulness on the least *recent* labels as with the DisTEMIST languages. The physicians were tasked with ranking patients according to their relevance to describe a diagnosis. Remarkably, both models rank the patients similarly, i.e., they successfully identify *prototypical* patients. In terms of the qualitative differences of the saliencies, doctors find that S-PROTO[XLM-R]is more sensitive to learning *annotation artifacts* (Zellers et al., 2019) like stopwords or explicit mentions of acronyms of the diagnosis, explaining

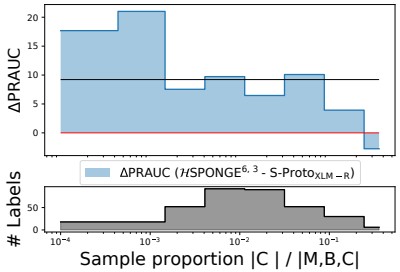 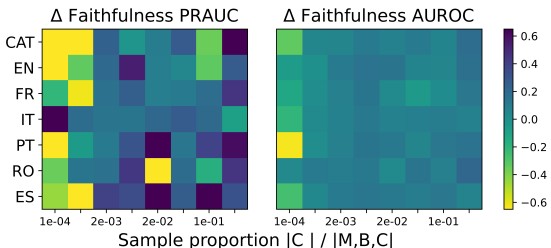

Figure 4: **Left** *top*: $\Delta(\mathcal{H}\mathrm{SPONGE}^{6,3} - \mathrm{S\text{-}PROTO}_{\mathrm{XLM\text{-}R}})$ in PRAUC over label *recency*. *Bottom*: number of labels by *recency* group. On average (gray dotted line) $\mathcal{H}\mathrm{SPONGE}^{6,3}$ generalizes better for all label groups. **Right** $\Delta(\mathcal{H}\mathrm{SPONGE}^{6,3} - \mathrm{S\text{-}PROTO}_{\mathrm{XLM\text{-}R}})$ Difference in faithfulness for zero-shot datasets (lower is better). Although faithfulness is similar for both models ($\Delta$ is roughly 0), $\mathrm{S\text{-}PROTO}_{\mathrm{XLM\text{-}R}}$ is less explainable for the under-represented labels in the last fine-tuning (most negative left region).

its weakness in generalization to distinct languages. This issue is compounded by $\mathrm{S\text{-}PROTO}_{\mathrm{XLM\text{-}R}}$ assigning near-uniform relevance scores to most tokens. In contrast, $\mathcal{H}\mathrm{SPONGE}^{6,3}$ sharply highlights key terms related to each diagnosis. These qualitative findings suggest that the zero-shot generalization capabilities of $\mathcal{H}\mathrm{SPONGE}^{6,3}$ is underpinned by clinically meaningful and explainable saliencies.

# 9 Conclusion

We address the challenges of sequential transfer learning when data is subject to privacy constraints by the example of the clinical domain. We focus on a multi-step sequential transfer learning setting, transferring knowledge via model checkpoints rather than raw data. We propose SPONGE, a novel architecture that excels at knowledge transfer and achieves superior performance in diagnoses prediction. We generalize *winner-takes-all*, a mechanism of neural inhibition inspired by nature, creating competing representations. Extensive analysis reveals that these representations combined with the ability of $\mathcal{H}$ydra to mitigate label-recency bias, are key to strong generalization performance. We confirm with medical doctors that the explanations provided by SPONGE are supported by clinical relevance.

**Future Work**. We examine a set of six datasets in nine different Indo-European languages, in multi-step sequential transfer learning. While this evaluation is significant, it is not exhaustive; further analysis will be conducted as more clinical data becomes available. Although we focus on a multi-label classification task, Transformers are widely used for solving generative tasks. Given that our architecture may also enhance decoder-based Transformer models, future evaluation on generative tasks is warranted as LLMs become more computationally efficient.

**Deployment considerations**. In our evaluation, SPONGE performs strongly across high- and low-resource data, sequential training, and generalization tasks. However, the scores achieved still leave room for improvement, which is crucial for the clinical domain. We emphasize that this work does not aim to replace doctors; the applications of technologies to the clinical environment must meet stringent legal and ethical requirements.

# 10 Acknowledgements

We thank the reviewers and the Action Editor for their constructive feedback, which improved this work. This work is funded by the Deutsche Forschungsgemeinschaft (DFG, German Research Foundation) Project-ID 528483508 - FIP 12, as well as the European Union under the grant project 101079894 (COMFORT - Improving Urologic Cancer Care with Artificial Intelligence Solutions), as well as by BMWe SOOFI, Grant Agreement 13IPC040D. The views expressed are solely those of the authors and do not necessarily reflect those of the European Union or European Health and Digital Executive Agency (HaDEA); neither is responsible for them.

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

## A Datasets

We present in Table 5 all data splits created with stratified sampling for the datasets used for training and evaluation in our experiments. DisTEMIST comprehends the same 439 EHR notes in seven different languages, for a total of 3,073 samples.

## B Impact of sequential training on downstream performance

We compare the average performance of models trained sequentially against models trained exclusively on a single dataset. Table 6 shows the absolute performance difference for every dataset. With very few exceptions, the performance boost is significant, highlighting the positive effect of knowledge transfer in sequential training. SPONGE[6,3] outperforms all models on all datasets on the average sequence when finetuned sequentially. S-Proto$_{\text{XLM-R}}$ outperforms XLM-R and XLM-R + A when trained and evaluated only on MIMIC(single dataset). However, for the other datasets that are trained in isolation S-Proto$_{\text{XLM-R}}$ is the

Table 5: Dataset splits.

| Dataset | Train | Val | Test |
|---|---|---|---|
| (M) MIMIC (Johnson et al., 2023) | 102,199 | 9,358 | 7,618 |
| (A) Achepa (Papaioannou et al., 2022) | 1,590 | 407 | 394 |
| (S) Stockholm University (Lamproudis et al., 2023) | 1,583 | 256 | 232 |
| (C) CodieEsp (Miranda-Escalada et al., 2020) | 656 | 158 | 182 |
| (B) SemclinBr (Oliveira et al., 2022) | 453 | 107 | 109 |
| (D) DisTEMIST (Miranda-Escalada et al., 2022) | — | — | 3,073 (439 × 7) |

worst-performing model, indicating that it is not suitable for low-resource datasets. Moreover, SPONGE[6,3] outperforms all models also for the case of single datasets which shows the data efficiency of our approach.

Generally, all models perform better when sequentially trained, this is confirmed by the positive performance difference for both metrics (bottom of Table 6).

Table 6: Performance difference when comparing sequential training results versus single-dataset training for every downstream dataset. We underline the best results when training on a single dataset (top), and **boldface** (middle) stands for the best results in sequential-transfer learning. All models benefit significantly(bottom) from multi-step sequential transfer learning.

| | | macro AUROC | | | | | macro PRAUC | | | | |
|---|---|---|---|---|---|---|---|---|---|---|---|
| | Model/Dataset | M | A | B | C | S | M | A | B | C | S |
| Single dataset | S-Proto$_{XLM-R}$ | 87.51 | 78.70 | 56.55 | 65.46 | 69.29 | 28.08 | 34.41 | 7.17 | 13.38 | 26.73 |
| | SPONGE[6,3] | 89.70 | 89.48 | 67.05 | 82.92 | 84.17 | 34.17 | 60.65 | 20.53 | 36.32 | 43.33 |
| | XLM-R | 86.21 | 87.37 | 65.18 | 77.74 | 87.01 | 27.27 | 55.80 | 18.29 | 28.18 | 63.80 |
| | XLM-R+A | 85.74 | 81.43 | 63.68 | 79.69 | 87.35 | 26.38 | 52.08 | 16.73 | 31.54 | 53.12 |
| Sequence average | S-Proto$_{XLM-R}$ | - | 81.83 | 70.75 | 83.33 | 84.42 | - | 42.22 | 23.83 | 34.66 | 47.33 |
| | SPONGE[6,3] | - | **90.05** | **82.65** | **91.03** | **90.62** | - | **59.93** | **42.16** | **55.22** | **59.67** |
| | XLM-R | - | 85.78 | 73.30 | 85.67 | 87.63 | - | 58.85 | 27.73 | 39.20 | 57.90 |
| | XLM-R+A | - | 84.83 | 75.93 | 86.38 | 88.08 | - | 55.65 | 28.54 | 38.77 | 57.52 |
| Performance Difference | S-Proto$_{XLM-R}$ | - | +3.13 | +14.20 | +17.87 | +15.13 | - | +7.81 | +16.66 | +21.28 | +20.60 |
| | SPONGE[6,3] | - | +0.57 | +15.6 | +8.11 | +6.45 | - | -0.72 | +21.63 | +18.9 | +16.34 |
| | XLM-R | - | -1.59 | +8.12 | +7.93 | +0.62 | - | +3.05 | +9.44 | +11.02 | -5.90 |
| | XLM-R+A | - | +3.4 | +12.25 | +6.69 | +0.73 | - | +3.57 | +11.81 | +7.23 | +4.4 |

## C  Hyperparameters

We use subnetwork exploration for 10 epochs to build distributed knowledge of MIMIC, with a batch size of 8 and learning rate as per Pfeiffer et al. (2020). We disable exploration for the remaining datasets of the training sequences. For S-Proto$_{XLM-R}$ we use hyperparameters as in Figueroa et al. (2024). SPONGE and S-Proto$_{XLM-R}$ use the same hidden dimensions for the prototypical layer, XLM-R as a Transformer, and are trained on four A100 GPUs.

**Adapter choice**  We evaluate SPONGE[1,1] and XLM-R+A across the four low-resource datasets using different adapter types: prefix-tuning (Li & Liang, 2021), LoRA (Hu et al., 2022), prompt-tuning (Lester et al., 2021) and bottleneck adapters (Pfeiffer et al., 2020). Consistent with findings from Poth et al. (2023), bottleneck adapters are the most effective for downstream classification tasks for SPONGE. For XLM-R, LoRA delivered the best performance. We use the best performing adapters for each model family. All models with adapters have significantly less trainable parameters than both S-Proto$_{XLM-R}$ and XLM-R.

**Cross-lingual Transformer choice.**  Because of the large amount of experiments and the extent of our evaluation, we consider smaller Transformers like XLM-R instead of significantly larger LLMs. We favor XLM-R over Glot500 (Imani et al., 2023) since the latter performs worse on text classification tasks in all but one of the languages of our training datasets (see Table 19 in Imani et al. (2023)).

## D  Best performing sequence checkpoint of S-Proto for DisTEMIST

In Table 7 we report the best performing sequence results of S-PROTO$_{\text{XLM-R}}$ and compare them to those of $\mathcal{H}$SPONGE$^{6,3}$. This sequence corresponds to Mimic(English) $\rightarrow$ SemClinBr(Portuguese) $\rightarrow$ CodiEsp(Spanish) or M$\rightarrow$B$\rightarrow$C. The two models resulting from this sequence significantly outperform the average scores over all sequences of all models reported in Table 3.

Table 7: Sequence $M \rightarrow B \rightarrow C$ yields the best S-Proto model for DisTEMIST, we report results for this sequence for $\mathcal{H}$SPONGE$^{6,3}$.

| Model | macro AUROC | micro AUROC | macro PRAUC |
|---|---|---|---|
| S-Proto | 86.16 | 84.73 | 25.25 |
| $\mathcal{H}$SPONGE$^{6,3}$ | 92.26 | 91.89 | 34.19 |

## E  All training sequences

In Figure 5 we list all generated 64 dataset sequences that start with MIMIC. Every node corresponds to the end of a sequence and a model checkpoint.

## F  Trainable parameters

Table 8 demonstrates the number of trainable parameters of the models in our experiments. We decompose these according to the corresponding modules of the architectures. Although SPONGE$^{6,3}$ features the largest number of competing subnetworks (90) it is the second smallest model w.r.t. the number of parameters that are updated. Note, that $\mathcal{H}$ydra is used only for inference and involves no training parameters. In our largest model and for our longest sequence of datasets, $\mathcal{H}$ydra adds roughly 1.8% parameters when compared to XLM-R, presenting no significant overhead.

Table 8: Number of subnetworks and trainable parameters for each model

| Model | Subnetworks | Transformer | Adapters | Projection $L$ | Prototypical parameters ($W$ and $u$) | Total |
|---|---|---|---|---|---|---|
| S-PROTO$_{\text{XLM-R}}$ | 1$\times$5 | 270M | - | 0.19M | 5$\times$0.25M | 271.5M |
| XLM-R | 1 | 270M | - | | - | 270M |
| SPONGE$^{6,3}$ | 6$\times$3$\times$5 | - | 6$\times$0.88M | 3$\times$0.19M | 5$\times$0.25M | 7.15M |
| SPONGE$^{6,1}$ | 6$\times$1$\times$5 | - | 6$\times$0.88M | 1$\times$0.19M | 5$\times$0.25M | 6.72M |
| SPONGE$^{1,6}$ | 1$\times$6$\times$5 | - | 1$\times$0.88M | 6$\times$0.19M | 5$\times$0.25M | 5.99M |
| SPONGE$^{1,1}$ | 1$\times$1$\times$5 | - | 1$\times$0.88M | 1$\times$0.19M | 5$\times$0.25M | 2.32M |
| XLM-R+A | 1 | - | 0.88M | | - | 0.88M |

## G  Experiments in a transfer learning scenario using same language datasets

Although the nature of clinical data is decentralized and often cross-lingual, we argue that sparse representations can also be beneficial for knowledge transfer in standard monolingual settings. To evaluate this, we conduct the same sequential transfer learning experiments, for all the sequences of datasets as in Figure 5, however, with English translations. To translate the datasets we use EuroLLM (Martins et al., 2025) since it is an open source SOTA model that can run locally while preserving the privacy of the clinical records. In addition, we focus on biomedical encoders that are pretrained on English corpora, and are SOTA in multi-label classification for English clinical text (Grundmann et al., 2025), namely, BiomedBERT (Gu et al., 2022) and ClinicalModernBERT (Lee et al., 2025). Following the approach described in Section 6 we report the performance metrics in Table 9 for the monolingual setting.

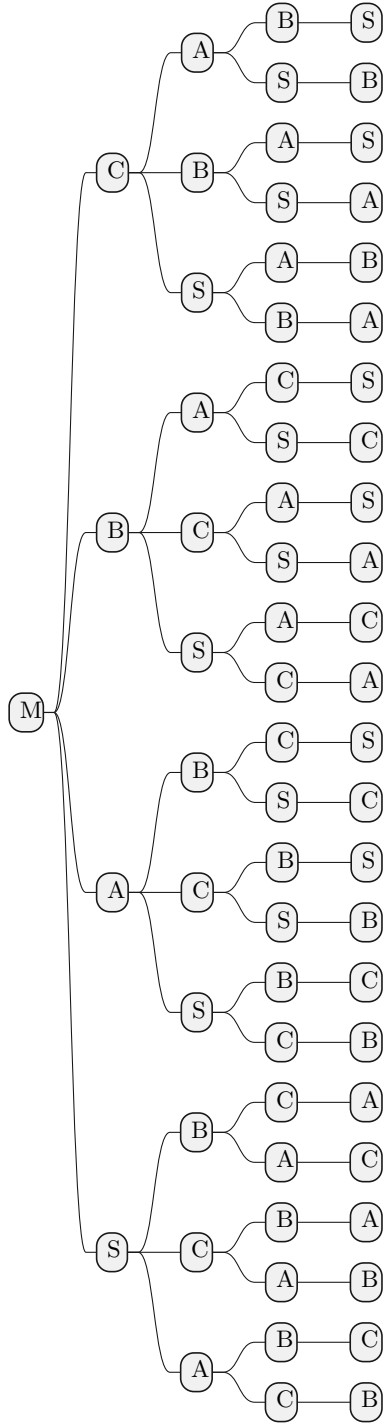

Figure 5: All 64 training sequences that we evaluate in our experiments.

From the two baselines BiomedBERT performs better, hence we create versions of our most sparse model $\text{SPONGE}^{6,3}_{\text{BiomedBERT}}$, as well as, S-PROTO$_{\text{BiomedBERT}}$ with this Transformer encoder. We add to the results, the best performing model $\text{SPONGE}^{6,3}_{\text{XLM-R}}$ from the cross-lingual setting (see Table 1) for reference.

We highlight that all models exhibit better performance for the monolingual experiments when compared to the cross-lingual setting. This could be attributed to the specialized pre-training of BiomedBERT and its domain specific tokenizer. Nevertheless, consistent with the cross-lingual results of Table 1, $\text{SPONGE}^{6,3}$ outperforms the other methods, suggesting that SPONGE can also boost performance in a standard monolingual transfer learning setting. When focusing only on the clinical outcome prediction task (van Aken et al., 2021) on the MIMIC dataset (left of Table 9), we underline how the cross-lingual $\text{SPONGE}^{6,3}_{\text{XLM-R}}$ is still a very competitive method even when compared to domain specialized models for English clinical text.

Table 9: Results of experiments in a monolingual (English) transfer learning scenario. (Left) model performance on the MIMIC dataset. (Right) average performance on the last dataset of each sequence. $\text{SPONGE}^{6,3}$ consistently outperforms across both settings.

| Model | MIMIC Only | | | Sequential Training | | |
|---|---|---|---|---|---|---|
| | macro AUROC | micro AUROC | macro PRAUC | macro AUROC | micro AUROC | macro PRAUC |
| BiomedBERT | 87.73 | 93.84 | 31.42 | 91.01 | 93.15 | 60.20 |
| ClinicalModernBERT | 86.77 | 93.88 | 29.79 | 86.72 | 90.67 | 51.64 |
| S-PROTO$_{\text{BiomedBERT}}$ | 88.73 | 94.42 | 35.83 | 91.21 | 93.50 | 58.91 |
| $\text{SPONGE}^{6,3}_{\text{BiomedBERT}}$ | **90.45** | **95.08** | **39.51** | **92.63** | **94.64** | **66.11** |
| $\text{SPONGE}^{6,3}_{\text{XLM-R}}$ | 89.70 | 94.70 | 34.17 | 88.58 | 92.09 | 54.24 |

