# OpenReview forum: "SPONGE: Competing Sparse Language Representations for Effective Knowledge Transfer"
_TMLR — Accepted by TMLR_

### Review · Reviewer_XPbz · 2025-06-09

**Summary Of Contributions:**

This paper presents a method for diagnosis prediction from electronic health records (EHR) via multi-step sequential transfer learning. The proposed method, SPONGE, incorporates multiple task adapters and a prototypical network to enable diagnosis prediction with a winner-takes-all strategy among subnetworks. The architecture is parameter-efficient and further boosted by Hydra, a method to overcome “label recency”, where consecutively fine-tuned models tend to favor the most recent set of labels encountered. Models are evaluated on multiple fine-tuning sequences across datasets that vary in terms of language, label sets, imbalance, etc. Results show that the sparsest form of SPONGE outperforms other approaches in virtually all scenarios, Hydra improves performance in the zero-shot setting, and an early “exploration” phase in subnetwork selection provides an additional performance boost.

**Audience:**

Yes

**Broader Impact Concerns:**

I don’t personally believe that a Broader Impacts statement is necessary for this submission.

**Claims And Evidence:**

Yes

**Requested Changes:**

**Main Feedback**
- If possible, include additional baseline methods for benchmarking. If not possible or prohibitive, please explain why.
- Please add an experiment evaluating the proposed methods in a multi-step sequential transfer learning setting where language does *not* change.
- Please clarify the “Analysis of saliencies with doctors.” Ideally, the methodology for this analysis should be described earlier in the Methods section. My main issue is that this paragraph does not provide any concrete results as far as I can tell. This paragraph does not reference a figure or table and seems to vaguely discuss the results of an experiment that I cannot find elsewhere in the paper. As written, I would simply remove this analysis and its corresponding sentence from the Abstract unless it can be clarified and supported with concrete experimental evidence.
- Related Work: The first paragraph can be bolstered with more relevant background and related approaches on disease prediction from clinical text [1-5]. This is such a well-studied area that perhaps a summary/review paper would be useful to reference as well.

**Minor Feedback**
- Top of page 2: I might suggest semicolons to separate the numbered items
- Section 4: Change “MIMIC – (High-resource)” to “MIMIC (High-resource)”? Unsure why the dash is included.
- Experiments: I don’t understand what XLM-R + A means without further explanation.
- Results: Change “This is evident when comparing SPONGE$^{6,1}$, SPONGE$^{1,6}$ against…” -> “This is evident when comparing SPONGE$^{6,1}$ and SPONGE$^{1,6}$ against…”
- In Tables, I would write “XLM-R” for consistency with the text.
- For Appendix F, also include the number of trainable parameters for all other SPONGE variants.

**References**

[1] He, Kai, et al. "A survey of large language models for healthcare: from data, technology, and applications to accountability and ethics." Information Fusion (2025): 102963.

[2] Rasmy, Laila, et al. "Med-BERT: pretrained contextualized embeddings on large-scale structured electronic health records for disease prediction." NPJ digital medicine 4.1 (2021): 86.

[3] Yang, Zhichao, et al. "Knowledge injected prompt based fine-tuning for multi-label few-shot icd coding." Proceedings of the conference on empirical methods in natural language processing. Conference on empirical methods in natural language processing. Vol. 2022. 2022.

[4] Liu, Yang, et al. "Effective convolutional attention network for multi-label clinical document classification." Proceedings of the 2021 Conference on Empirical Methods in Natural Language Processing. 2021.

[5] Zhang, Shurui, et al. "Automatic ICD coding exploiting discourse structure and reconciled code embeddings." Proceedings of the 29th International Conference on Computational Linguistics. 2022.

**Strengths And Weaknesses:**

**Strengths**:
- The paper is generally well-written and organized clearly. There is a logical flow from one passage to the next, and illustrations help guide the reader through the narrative of the study.
- The problem setting is explained well and properly motivated: in highly sensitive spaces like healthcare, it is often preferable to share models (weights) than data. This poses unique challenges that the paper then identifies and crafts a method to solve.
- The release of open-source code to enable reproducibility is appreciated.
- The experimental framework is unique and well-suited to the domain. Experiments are thorough and appear to be soundly conducted. There are many relevant experiments and ablation studies showing the value of the proposed approach.

**Weaknesses**:
- I would be interested in comparison to other baseline approaches. I am not intimately familiar with the space but I would imagine there are multiple recent efforts for clinical text understanding that might be competitive in diagnosis prediction other than S-Proto and XLM-R
- I am left wondering whether the results and takeaways are driven by the cross-lingual nature of this specific experimental setup. In other words, is SPONGE/Hydra still useful (or as useful) in a *same-language* sequential transfer learning framework? This is a common (and still challenging) setting, where differences in patient demographics, labels, and many other factors can differ from dataset to dataset. If the authors could present some auxiliary results in a setting like this, then the practical applicability of this method might be improved – i.e., it is not always necessary to train a model with multi-lingual capabilities.

---

> ### Author Response · Authors · 2025-11-06
> **Response to your review**
>
> We thank the reviewer for their valuable feedback.
>
> > If possible, include additional baseline methods for benchmarking. If not possible or prohibitive, please explain why.
> > Please add an experiment evaluating the proposed methods in a multi-step sequential transfer learning setting where language does not change.
>
> Thank you for these insightful points. For this we had to conduct several experiments again, but instead of using the original multilingual setting, we used the corresponding English translations. We used EuroLLM \[1\] as it captures the languages of our datasets best.
>
> In addition we added another recent competitive baseline, ClinicalModernBERT \[2\] in addition to BiomedBERT\[3\].
> The results make our claims in the paper even stronger, since our architecture outperforms all these competitive baselines. Furthermore, we  add the reference to CliniBench\[4\], a very recent study where they conduct a very thorough comparison of Transformer-encoder based methods with Generative Transformers (LLMs) on the MIMIC code prediction task. Here they highlight the strong performance of encoder based architectures, such as BiomedBERT and ClinicalModernBERT.
>
>
> A full presentation of these results in the manuscript would technically double the amount of tables, thus, we decided to add a summary in the appendix (Section G) with the most important results:
>
> For the initial step in the sequence (MIMIC IV)
>
> | Model                  | macro AUROC | micro AUROC | macro PRAUC |
> |------------------------|-------------|-------------|-------------|
> | BiomedBERT             | 87.73       | 93.84       |  31.42      |
> | ClinicalModernBERT     | 86.77       | 93.88       | 29.79       |
> | S-Proto_BiomedBERT     | 88.73       | 94.42       | 35.83       |
> | SPONGE(6,3)\_BiomedBERT | **90.45**      | **95.08**       | **39.51**       |
>
> For all the 64 sequences:
>
> | Model                  | macro AUROC | micro AUROC | macro PRAUC |
> |------------------------|-------------|-------------|-------------|
> | BiomedBERT             | 91.01       | 93.15       | 60.20       |
> | ClinicalModernBERT     | 86.72       | 90.67       | 51.64       |
> | S-Proto_BiomedBERT     | 91.21       | 93.50       | 58.91       |
> | SPONGE(6,3)\_BiomedBERT | **92.63**       | **94.64**       | **66.11**       |
>
> \[1\] Martins, P. H., Alves, J., Fernandes, P., Guerreiro, N. M., Rei, R., Farajian, A., ... & Martins, A. F. (2025). Eurollm-9b: Technical report. arXiv preprint arXiv:2506.04079.
>
> \[2\] Lee, S. A., Wu, A., & Chiang, J. N. (2025). Clinical modernbert: An efficient and long context encoder for biomedical text. arXiv preprint arXiv:2504.03964.
>
> \[3\]  Chakraborty, S., Bisong, E., Bhatt, S., Wagner, T., Elliott, R., & Mosconi, F. (2020, December). BioMedBERT: A pre-trained biomedical language model for QA and IR. In Proceedings of the 28th international conference on computational linguistics (pp. 669-679).
>
> \[4\] Grundmann, P., Fast, D., Frick, J., Steffek, T., Gers, F., Nejdl, W., & Löser, A. (2025). CliniBench: A Clinical Outcome Prediction Benchmark for Generative and Encoder-Based Language Models. arXiv preprint arXiv:2509.26136.
>
> > Please clarify the “Analysis of saliencies with doctors.” Ideally, the methodology for this analysis should be described earlier in the Methods section.
>
> Thank you for this remark. In the qualitative evaluation with doctors, the comparison of saliencies is between S-Proto and SPONGE and is based on the following questions:
>
> We chose thirty clinical notes belonging to five different diagnoses where both models predicted correctly, and had the smallest distances to the learned prototypical vectors.  We presented six saliency maps of the five diagnoses (weighting the text) where the doctors do not know which belongs to what model. And ask them the following questions:
>
> 1) Are the patient notes characteristic for a specific diagnosis? Do they encompass a prototypical patient.
>
> 2) Are the highlighted terms relevant for the predicted diagnoses?
>
> We focused on only one of the 65 checkpoints, the same as in the faithfulness analysis (Sec. 8 Fig. 4), and addressed a zero-shot language (Greek). Since a very thorough analysis on all checkpoints and languages is very costly in terms of the medical support we would need, we focus only on a qualitative survey, and expressed the doctors' opinion in our results.
>
> We will add more clarification regarding this into the manuscript, but we are also open to remove this section given reviewer j6HD also gave his valuable feedback regarding this.
>
> > Related Work: The first paragraph can be bolstered with more relevant background and related approaches on disease prediction from clinical text \[1-5\].
>
> Thank you for your thorough input on this, we have added these citations to the manuscript as well as their contributions in our related work.
>
> > Minor Feedback
>
> Thank your for this, we included all these recommendations in the manuscript.

---

### Review · Reviewer_Xo5B · 2025-09-04

**Summary Of Contributions:**

This paper addresses a key challenge in clinical natural language processing (NLP): how to effectively transfer knowledge between models when data is isolated due to privacy constraints. The standard approach is sequence transfer learning, which often fails because new knowledge overwrites previously learned information—a problem known as "catastrophic forgetting." To address this, this paper proposes a two-part solution: the SPONGE architecture, which prevents knowledge overwriting, and the Hydra strategy, which addresses the label recency bias problem. Key experimental results confirm that the SPONGE architecture is highly effective at sequential learning without forgetting, and when combined with the Hydra strategy, achieves state-of-the-art performance in generalizing to completely unseen data while being extremely parameter-efficient.

**Audience:**

Yes

**Claims And Evidence:**

Yes

**Requested Changes:**

See the disadvantage part.

**Strengths And Weaknesses:**

## Advantages

SPONSE, a novel architectural design, successfully integrates three distinct technical concepts—PEFT adapters, prototype networks, and a generalized WTA mechanism—into a cohesive whole. This innovation effectively addresses the core pain point of inter-task interference in multi-step sequence learning.

The Hydra strategy is integrated with SPONSE. Hydra is a simple, low-cost module applied only at inference time that accurately addresses the identified problem without increasing training complexity.

Compared to testing on a single, fixed curriculum, this paper employs a large-scale, sequence-independent evaluation approach. This approach provides a more realistic and robust assessment of model performance, making the conclusions more convincing and generalizable.

SPONGE achieves state-of-the-art performance while requiring only approximately 2.4% of the model's total parameters to train, a significant practical advantage. It significantly reduces the cost of training and adapting large models.

The model not only performs well across metrics, but also generates interpretable outputs recognized by domain experts. This is an important step in promoting the application of AI technology in high-risk fields such as healthcare.

## Disadvantages

While the Hydra strategy offers zero training overhead, it introduces significant complexity during inference. It requires storing and loading multiple sets of historical prototype parameters in memory. For very long learning sequences, this increases memory usage and computational latency during inference.

The WTA mechanism selects one subnetwork as the winner, while all other subnetworks are ignored. If the WTA selection process is affected by noise or makes a suboptimal choice for a particular input, the system becomes completely dependent on this suboptimal subnetwork, lacking the ability to correct errors or integrate signals from other "near winners."

The evaluation scope is limited to Indo-European languages and a single diagnostic and predictive task. No effort has been made to include more complex languages, such as Chinese.

The entire framework was designed and evaluated for classification tasks. No effort has been made to apply the entire framework to generative tasks to evaluate model performance.

---

> ### Author Response · Authors · 2025-11-06
> **Response to your review**
>
> We thank the reviewer for their valuable feedback.
>
> > While the Hydra strategy offers zero training overhead, it introduces significant complexity during inference. It requires storing and loading multiple sets of historical prototype parameters in memory. For very long learning sequences, this increases memory usage and computational latency during inference.
>
> Thank you for this question, in Appendix F, and Table 8 we higlight the training parameters of the prototypical network of SPONGE. Even though Hydra is only an inference strategy, for our longest sequence of 5 datasets the  total number of parameters that are added are $4\\times5\\times0.25$M parameters, or approximately 1.8% of the parameters of XLMR, which is a relatively small overhead when compared to the Transformer size. A sequence that would render a measurable  slowdown of inference (reach at least XLMR size) would involve  approximately 277 datasets, at this point different constraints might already dominate performance.
> Furthermore, note that these additional parameters are only used in only half of the inference step, before WTA has chosen a single subnetwork.
> We added this clarification in the Appendix F.
>
> > The WTA mechanism selects one subnetwork as the winner, while all other subnetworks are ignored. If the WTA selection process is affected by noise or makes a suboptimal choice for a particular input, the system becomes completely dependent on this suboptimal subnetwork, lacking the ability to correct errors or integrate signals from other "near winners."
>
> Thank you for this insightful comment, in Section 8 we  detail a survey on the optimality of Winner-takes-all, and show empirically, that although the strategy is not strictly optimal, it captures the order and thus correlation between the activation and performance of all subnetworks. The effect of Winner-Takes-All, was surveyed for this task in S-Proto\[1\] (Chapter 7, Figure 4), where they attribute the gains in performance to the random initialization and non-interfering learned prototypes.  We reinforce this explanation within the manuscript in Section 2.
>
> > The evaluation scope is limited to Indo-European languages and a single diagnostic and predictive task. No effort has been made to include more complex languages, such as Chinese.
>
> This is a valid remark, and we tried to the best of our effort to gather a wide variety of datasets in different languages. We train our sequences on up to five distinct languages and evaluate on up to 9. We acknowledge this as a scope limitation and we added this as part of our future work.
>
> > The entire framework was designed and evaluated for classification tasks. No effort has been made to apply the entire framework to generative tasks to evaluate model performance.
>
> We designed our experiments for classification tasks based on evidence showing that encoder-based architectures e.g., BiomedBERT outperform generative Transformers (LLMs) on clinical code prediction benchmarks like MIMIC, while being more computationally efficient \[2\]. We acknowledge this as a scope limitation and have added exploration of generative approaches as valuable future work in the  manuscript.
>
> \[1\] Figueroa, A., Papaioannou, J. M., Fallon, C., Bekiaridou, A., Bressem, K., Zanos, S., Gers, F. and Nejdl, W. & Löser, A. (2024, August). Boosting Long-Tail Data Classification with Sparse Prototypical Networks. In Joint European Conference on Machine Learning and Knowledge Discovery in Databases (pp. 434-449). Cham: Springer Nature Switzerland.
>
> \[2\] Grundmann, P., Fast, D., Frick, J., Steffek, T., Gers, F., Nejdl, W., & Löser, A. (2025). CliniBench: A     Clinical Outcome Prediction Benchmark for Generative and Encoder-Based Language Models. arXiv preprint arXiv:2509.26136.

---

### Review · Reviewer_j6HD · 2025-10-21

**Summary Of Contributions:**

The paper introduces a method to train a model to combine knowledge across datasets that can only be accessed in sequence, a procedure they explain to be realistic when sharing data via their model across institutions. They construct a novel evaluation procedure by combining existing datasets to evaluate their work and achieve good results on this benchmark.

Disclaimer: I am no expert in natural language processing for clinical data. As such, my review focuses on clarity and methodology.

**Audience:**

No

**Broader Impact Concerns:**

As the proposed model takes no measures to avoid memorizing the data, the "privacy constraint" seems not to be respected without further measures, making the model vulnerable to potential extraction attacks (A). The authors should comment on this
issue.

(A): Carlini, N., Tramer, F., Wallace, E., Jagielski, M., Herbert-Voss, A., Lee, K., ... & Raffel, C. (2021). Extracting training data from large language models. In 30th USENIX security symposium (USENIX Security 21) (pp. 2633-2650).

**Claims And Evidence:**

No

**Requested Changes:**

## Required
While these changes are required, the paper will look very different if they are adopted. I cannot guarantee that those will be enough for me to recommend acceptance. Further reviews might be required.

[R1] Overhaul the structure of the paper:
- Add a structure section at the end of the introduction.
- Separate introduction, methods, and related work. The introduction should describe the problem at hand and sketch your solution approach. Related work should focus on preliminary work and may list its limitations, but your method should be described in the method section.

[R2] Explain your architecture. While you depict the architecture in Figure 2, it is almost not used to aid the explanations in 5.1.

[R3] Properly introduce and explain the formulas in the method section (see [W5]).

## Minor

[M1] p.12 The citation (Zellers, et al., 2019) confused me, as it is placed in such a way as if it would support your claim, when its purpose is to explain "annotation artifacts".

[M2] Change the names/ title, see ([W4]).

[M3] Some acronyms are not introduced, ie. ICD-10, CCSR. The link to CCSR should be spelled out

**Strengths And Weaknesses:**

## Strengths
[S1] Evaluation Process: Constructing an evaluation procedure that closely resembles realistic conditions seems important.

[S2] Empirical Performance: The empirical performance seems great over a variety of tasks/ datasets.

## Weaknesses

[W1] Structure: The paper has a loose structure only. As a result, judging what is an original contribution and what is taken from prior work is hard for the reader. For example, the method section was not enough for me to understand the proposed method, as details are spread over the main text, cf. the related work section, i.e., p.4 "we use a parametric inhibition approach".

[W2] Related Work: Although there is a related work section, this section is very short for a 12-page manuscript, but it uses almost half of this space to introduce parts of the proposed method. Additional related work and concepts are introduced throughout the paper. This contributes to the difficulty of separating the original contribution. I.e., p.8 Experiments, p.5 Methods.

[W3] Reproducability: The paper's main contribution is a clinical data model, which achieves great scores on the newly constructed evaluation procedure. Reproducing those results based on the manuscript seems impossible. While parameter counts and the general architecture are provided, no concrete specifics on the parts are given. Appendix C delegates almost all hyperparameters for the proposed method to prior work, making the original contribution questionable.

[W4] Naming: While SPONGE and Hydra are catchy names, there is no justification for them. As a result, it just introduces additional overhead when reading. I prefer not to put a strong emphasis on a name - putting it in the title also does not help a reader judge whether this paper is relevant to them.

[W5] The little math used in the paper is inconsistent:
- p.6: The adapter $\eta^t\in\mathbb{R}^{s\times h}$ is defined to depend in size on the position of the index in the index set of tokens $s\in E$. What is $\mathbb{R}^{E}$ meant to represent? Do you mean a one-hot encoding $\mathbb{R}^{|E|}$? Later, the inner product between $\eta^t\in\mathbb{R}^{s\times h}$ and $L^p$ (which is never defined) is of size $\mathbb{R}^{C\times E}$, which is inconsistent with the notation before.

[W6] Dissemination of experiments: The description of individual experiments is very short, which makes it hard to judge the results. In addition, it remains unclear which algorithmic choice has which impact on performance, as numerous changes are evaluated simultaneously. In addition, the results are not always aligned with the conclusion, e.g., p.11 "Hydra and label recency" claims S-Proto and SPONGE "underperform (wrt. what?) when not modified with Hydra", but Table 3 shows that S-Proto performs *worse* on unseen data/ languages when modified with Hydra.

[W7] Description of human evaluation: While it is certainly interesting to see how the models align with clinicians' judgments, that description on p.12 did not enable me to understand how these findings were collected, i.e., based on a multiple choice test, an interview, etc. This evaluation would have been particularly interesting if it had been carried out and described more systematically.

[W8] Motivation of choices: Most of the architectural decisions are based on intuition and "speculation", p.5. As such, the value of this work to adjacent fields might be limited, as they might not generalize.

---

> ### Author Response · Authors · 2025-11-06
> **Response to your review**
>
> > \[R1\] Overhaul the structure of the paper:
>
> Thank you for your valuable input. Since our findings target a domain specific NLP task, we follow the structure of the relevant publications of model architectures of this task\[1,2,3,4\]. Therefore, the related work section generally puts in context all the methodological contributions. i.e., explains the common and contrasting decisions of each approach. By following this, we incorporate intuition on the methods also in the related work.
>
> \[1\] van Aken, B., Papaioannou, J.-M., Mayrdorfer, M., Budde, K., Gers, F., & Loeser, A. (2021). Clinical outcome prediction from admission notes using self-supervised knowledge integration. In Proceedings of the 16th Conference of the European Chapter of the Association for Computational Linguistics: Main Volume (pp. 881–893). Association for Computational Linguistics. https://aclanthology.org/2021.eacl-main.75
>
> \[2\] Naik, A., Parasa, S., Feldman, S., Wang, L. L., & Hope, T. (2022). Literature-augmented clinical outcome prediction. In Findings of the Association for Computational Linguistics: NAACL 2022 (pp. 438–453). Association for Computational Linguistics. https://doi.org/10.18653/v1/2022.findings-naacl.34
>
> \[3\] van Aken, B., Papaioannou, J.-M., Naik, M., Eleftheriadis, G., Nejdl, W., Gers, F., & Loeser, A. (2022). This patient looks like that patient: Prototypical networks for interpretable diagnosis prediction from clinical text. In Proceedings of the 2nd Conference of the Asia-Pacific Chapter of the Association for Computational Linguistics and the 12th International Joint Conference on Natural Language Processing (Volume 1: Long Papers) (pp. 172–184). Association for Computational Linguistics. https://aclanthology.org/2022.aacl-main.14
>
> \[4\] Figueroa, A., Papaioannou, J.-M., Fallon, C., Bekiaridou, A., Bressem, K., Zanos, S., Gers, F., Nejdl, W., & Löser, A. (2024). Boosting long-tail data classification with sparse prototypical networks. In A. Bifet, J. Davis, T. Krilavičius, M. Kull, E. Ntoutsi, & I. Žliobaitė (Eds.), Machine learning and knowledge discovery in databases. Research track (pp. 434–449). Springer Nature Switzerland. https://doi.org/10.1007/978-3-031-70368-3_26
>
> > \[R2\] Explain your architecture.
>
> This is great feedback, thank you for this. We added suitable references in section 5.1 to the methods figure, namely: the expansion of $T$ Transformer representations with the help of task adapters, the dimensional expansion of $L$ and the winner-takes-all strategy to find the optimal subnetwork $\\xi$
>
> > \[R3\] Properly introduce and explain the formulas in the method section (see \[W5\]).
>
> Thank you for catching this, it helps us improve greatly our work.  We corrected the notation where relevant, namely.
>
> - We corrected  $\\eta^t \\in \\mathbb{R}^{E \\times h}$, and what it denotes is the Transformer representation for a single token given an adapter $t$. Where $E$ is the sequence length of the Transformer encoder. Therefore, we removed references to $s$ since it is implicit.
>
>  - We  added the definition of $L^p$ with it's dimensionality, and this corresponds to indexing $L$ on the first dimension by $p$.
>
> Furthermore, regarding your concerns about reproducibility of our work \[W3\], we made all source code and hyperparameters to reproduce our experiments openly available under the link in p.1. Sadly, given the time-frames of the review process, the anonymous link expired so we renewed this.
>
> Regarding naming of our method \[W4\], we choose SPONGE as neologism of SParsity and a metaphor of knowledge absorption and transfer, and Hydra is analogous to the accumulation of multiple output subnetworks (or heads).
>
> As for the dissemination of experiments \[W6\], although we acknowledge that our ablations are not exhaustive because of computational constraints, we conduct several ablations to the best of our effort for the most important algorithmic choices:
>
> In Section 6 and 7, Table 1, we show an ablation of adding sparsity to the Transformer backbone. In the same Table, we ablate the impact of subnetwork initialization with our proposed exploration method. In Table 2, we analyze confounding factors such as the number of training datasets, as well as the impact of the dataset language. Additionally, reviewer XPbz, pointed out a great ablation needed, with respect to languages, for which we added in Appendix G, experiment results where we limit ourselves to a single language. Additionally, thank you for your catch on our observation regarding prototypical classifiers, we corrected this in the manuscript to refer only to our method.

---

> > ### Author Response · Authors · 2025-11-06
> > **Response to your review (Continuation)**
> >
> > Regarding the qualitative evaluation with doctors\[W7\] the comparison of saliencies is between S-Proto and SPONGE. We chose thirty clinical notes belonging to five different diagnoses where both models predicted correctly, and had the smallest distances to the learned prototypical vectors. We presented six saliency maps of the five diagnoses (weighting the text) where the doctors do not know which belongs to what model. And ask them the following questions:
> >
> > 1. Are the patient notes characteristic for a specific diagnosis? Do they encompass a prototypical patient.
> > 2. Are the highlighted terms relevant for the predicted diagnoses?
> >
> > We focused on only one of the 65 checkpoints, the same as in the faithfulness analysis (Sec. 8 Fig. 4), and addressed a zero-shot language (Greek). Since a very thorough analysis on all checkpoints and languages is very costly in terms of the medical support we would need, we focus only on a qualitative survey, and expressed the doctors' opinion in our results.
> >
> > We will add more clarification regarding this into the manuscript, but we are also open to remove this section given reviewer XPbz also gave his valuable feedback regarding this.
> >
> > Regarding \[W8\] and the intuition behind our architectural choices, the core of our work centers on generalizing the work of S-Proto\[4\] and creating end-to-end sparse, prototypical representations. Technically we create multiple parallel Transformer representations which are only feasible with PEFT (adapters) and we expand the bottleneck $L$ in order to remove the "non-sparse" modules. Putting it all together is the generalization of winner-takes-all that has to span all of these architectural modifications (eq.2-5). We added clarification for this in the first paragraph of section 5.1.
> >
> > Additionally thank you for your minor revision suggestions, we incorporated all of them in the manuscript.
> >
> > Regarding your broader impact concerns, we focused on Transformer encoder architectures (BERT family) which are not generative (as per your suggested citation (A)), and for which the extraction of the training data, although not impossible, is much more difficult \[1,2,3\]. We added these cites where we motivate this point in the introduction.
> >
> > \[1\] Lehman, E., Jain, S., Pichotta, K., Goldberg, Y., and Wallace, B. C. 2021. Does BERT Pretrained on Clinical Notes Reveal Sensitive Data? In Annual Conference of the North American Chapter of the Association for Computational Linguistics, NAACL.
> >
> > \[2\] Vakili, T., & Dalianis, H. (2021). Are clinical BERT models privacy preserving? The difficulty of extracting patient-condition associations. In AAAI 2021 Fall Symposium on Human Partnership with Medical AI: Design, Operationalization, and Ethics (AAAI-HUMAN 2021), Virtual Event, November 4-6, 2021.
> >
> > \[3\] Huang, J., Shao, H., & Chang, K. C.-C. (2022). Are large pre-trained language models leaking your personal information? In Findings of the Association for Computational Linguistics: EMNLP 2022 (pp. 2038–2047). Association for Computational Linguistics.

---

### Decision · Action_Editor_j4f1 · 2025-11-26

**Recommendation:** Accept with minor revision

**Additional Comments:**

I would suggest a very slight revision to the scope clarification. (see the aforementioned discussions regarding this point)

**Audience:**

Yes

**Audience Explanation:**

The reviewers appreciated the practical utility of the proposed method and agreed with the implications of the paper.

**Claims And Evidence:**

Yes

**Claims Explanation:**

This paper addresses a key challenge in clinical natural language processing (NLP): how to effectively transfer knowledge between models when data is isolated due to privacy constraints. The standard approach is sequence transfer learning, which often fails because new knowledge overwrites previously learned information—a problem known as "catastrophic forgetting." To address this, this paper proposes a two-part solution: the SPONGE architecture, which prevents knowledge overwriting, and the Hydra strategy, which addresses the label recency bias problem. Key experimental results confirm that the SPONGE architecture is highly effective at sequential learning without forgetting, and when combined with the Hydra strategy, achieves state-of-the-art performance in generalizing to completely unseen data while being extremely parameter-efficient. However, they acknowledged certain limitations on the scope of applicability.

**The title and framing describe a general framework for sequential transfer learning, but all of the experiments involve cross-lingual domain gaps and in healthcare. How does the model perform in more standard transfer learning/fine-tuning settings, where a model is simply adapted to a new domain/task/etc. in the same language? The reviewer raised this point earlier, and it was not addressed by the authors. Therefore their suggestion would be to either address this with experiments or simply reign in the language (and even the title of the paper) to more specifically address that this is a method for cross-lingual knowledge transfer.**

I agree with this reviewer's evaluation. I would suggest a very slight revision to the scope clarification.

---

> ### Author Response · Authors · 2025-12-01
> **Answer to the action editor and revision**
>
> We thank the action editor and the reviewers for their valuable input.
> We agree with your feedback regarding narrowing the scope of the manuscript given our evaluation and experiments.
> Therefore, we have settled on an updated title in line with the cross-lingual aspect of the tasks and the clinical domain "SPONGE: Competing Sparse Language Representations for Effective Cross-Lingual Knowledge Transfer in Healthcare".
>
> Additionally, in section 7 (Results) p.9 we have polished the reference to Appendix G, where we clarify more thoroughly the  experiments we conducted with sequential training in the same language as per the input of reviewer XPbz.
> For more clarity we renamed and expanded Appendix G, discussing the results of these experiments.
> We highlight how our method also outperforms the other architectures in a more standard transfer learning setting in the same language, albeit on tasks in the same domain (clinical) with different label spaces.
> We have uploaded a new revision of the manuscript with these changes, please confirm whether this is in line with the final camera ready revision so that we can upload the deanonimized file.
> Thank you once again for your feedback and support.